

# Adaptive mechanisms in quinoa for coping in stressful environments: an update

Qura Tul Ain[1], Kiran Siddique[2], Sami Bawazeer[3], Iftikhar Ali[4,5], Maham Mazhar[1], Rabia Rasool[1], Bismillah Mubeen[1], Farman Ullah[6], Ahsanullah Unar[7] and Tassadaq Hussain Jafar[8]

[1] Institute of Molecular Biology and Biotechnology, University of Lahore, Lahore, Punjab, Pakistan
[2] School of Life Sciences and Biotechnology, Shanghai Jiao Tong University, Shanghai, China
[3] Faculty of Pharmacy, Department of Pharmacognosy, Umm Al-Qura University, Makkah, Makkah, Saudi Arabia
[4] Department of Genetics and Development, Columbia University, New York, United States
[5] Center for Plant Sciences and Biodiversity, University of Swat, Swat, Pakistan
[6] Center for Biotechnology and Microbiology, University of Swat, Swat, Pakistan
[7] School of Life Sciences, University of Science & Technology, China, Hefei, China
[8] University of Molise, Campobasso, Molise, Italy

Corresponding author
Tassadaq Hussain Jafar,
t.jafar@studenti.unimol.it

## ABSTRACT

Quinoa (*Chenopodium quinoa*) is a grain-like, genetically diverse, highly complex, nutritious, and stress-tolerant food that has been used in Andean Indigenous cultures for thousands of years. Over the past several decades, numerous nutraceutical and food companies are using quinoa because of its perceived health benefits. Seeds of quinoa have a superb balance of proteins, lipids, carbohydrates, saponins, vitamins, phenolics, minerals, phytoecdysteroids, glycine betaine, and betalains. Quinoa due to its high nutritional protein contents, minerals, secondary metabolites and lack of gluten, is used as the main food source worldwide. In upcoming years, the frequency of extreme events and climatic variations is projected to increase which will have an impact on reliable and safe production of food. Quinoa due to its high nutritional quality and adaptability has been suggested as a good candidate to offer increased food security in a world with increased climatic variations. Quinoa possesses an exceptional ability to grow and adapt in varied and contrasting environments, including drought, saline soil, cold, heat UV-B radiation, and heavy metals. Adaptations in salinity and drought are the most commonly studied stresses in quinoa and their genetic diversity associated with two stresses has been extensively elucidated. Because of the traditional wide-ranging cultivation area of quinoa, different quinoa cultivars are available that are specifically adapted for specific stress and with broad genetic variability. This review will give a brief overview of the various physiological, morphological and metabolic adaptations in response to several abiotic stresses.

## INTRODUCTION

It is expected that the human population will increase from six billion to nine billion in 2050 with an increase in food demand between 70 and 100% that strongly impacts reliable

food production (*Godfray et al., 2010*). It is estimated that about one in every eight individuals is suffering from undernourishment with the mounting prevalence of obesity, diabetes, osteoporosis, fragility, and other metabolic disorders (*Nguyen & Lau, 2012*; *Zimmet et al., 2014*). Food can play a significant role in disease prevention and treatment and serve as a strong integrative strategy to combat age-related disorders and metabolic diseases. Food that gives beneficial effects on humans is termed functional food and its effects range from reduction of disease risk to its specific treatment strategy (*Bigliardi & Galati, 2013*). It is estimated that about 30 to 70% of the human daily energy requirement depends upon cereal-based foods (*Poutanen, Sozer & Valle, 2014*).

Quinoa (*Chenopodium quinoa*) is a food crop that is increasingly used as a food source worldwide but has been used in Andean Indigenous cultures for thousands of years (*Alandia et al., 2020*). Quinoa is also known as a superfood because of its super nutritious quality and stress-tolerant properties. Quinoa is mistakenly considered pseudocereal because of its property similar to that of cereal grains like corn, wheat, and rice, however, quinoa is classified as a dicot (*Kadereit et al., 2003*). Quinoa is morphologically and systemically different from cereal grains because it belongs to the family Amaranthaceae with unique seed and fruit anatomy. Quinoa fruits are achenes with a single seed enclosed by the outer pericarp. A quinoa seed consists of central periplasm with localized carbohydrate reserves surrounded by circular protein-rich and oil-rich embryos with endosperm and enclosed by a seed coat (*Ruiz et al., 2014*). Quinoa seeds undergo de-saponification/de-husking *via* washing and mechanical abrasion to remove bitter saponins and leave the nutrient-rich embryo and endosperm intact. The leaves of quinoa have been consumed similarly to spinach and are also used in salads (*McKeown et al., 2013*). The quinoa seed has been consumed in many different ways such as an important component of soup, as breakfast cereals, or used as flour in baked and toasted goods (bread, noodles, biscuits, cookies, flakes, pancakes, and tortillas). The whole quinoa plant also has been used as a rich source of food for livestock such as pigs, poultry, and cattle (*Bhargava, Shukla & Ohri, 2006*).

Quinoa is a unique and culturally important stress-tolerant crop with different phytochemical compositions and high nutritional value. Quinoa has a unique micro- and macronutrient profile such as lipids, amino acids, and carbohydrates and its secondary metabolites may also contribute to health benefits (*Angeli et al., 2020*). The most important group of secondary metabolites reported in quinoa are phenolics, glycine betaine, betalains, triterpenoids (phytosterols, saponins, phytoecdysteroids), *etc*. The quality and quantity of protein in quinoa are better than those of other cereal grains with high digestibility and gluten-free property. Quinoa has higher total protein content as compared to other cereal grains such as barley, rice, maize, oats, *etc*. the storage proteins mostly consist of albumin and globulin and a minor amount of prolamins (*Sindhu & Khatkar, 2019*). An *in vitro* study on the presence of gluten in quinoa plants revealed that only two out of 15 cultivars showed celiac toxic prolamin epitopes and these results show that quinoa is gluten-free and safe for consumption. Chenopodin is a protein that constitutes about 37% of the total protein content in quinoa which is a rich source of isoleucine,

leucine, tyrosine, and phenylalanine recommended by Food and agriculture organization (FAO) (*Suárez-Estrella et al., 2018*).

Quinoa comprises 58.1 to 64.2% of starch and has a low glycemic index (*Sharma & Lakhawat, 2017*). The starch mainly consists of glucose with small amounts of maltose, D-xylose and fructose. The starch consists of small granules and is highly branched and smaller than the particle size of common cereal grains (*Vega-Gálvez et al., 2010*). Dietary fibers in quinoa plants are resistant to digestion by enzymes in the small intestine and even absorption and undergo partial fermentation in the large intestine. Dietary fiber can reduce the risk of infection and inflammation by reducing lipid and cholesterol absorption, improving microbiota in the intestine, and modulating postprandial insulin response (*Brownawell et al., 2012*; *De Carvalho et al., 2014*). Insoluble quinoa fiber is mainly composed of arabinose, xylose, galacturonic acid, and glucose that constitute about 78% of total fiber content while soluble fiber content such as arabinose, galacturonic acid, and glucose constitute about 22% of total fibers which is higher than that of maize and wheat (*Lamothe et al., 2015*).

Lipid content in quinoa is higher than that of maize and ranges from 2 to 10%. The composition of seed oil in the quinoa plant includes 54.2 to 58.3% polyunsaturated fatty acids and 89.4% unsaturated fatty acids. The ratio of polyunsaturated fatty acids is mostly 18:2n-6 and 18:3n-3 with a 6:1 ratio of omega-6 and omega-3 which is more favorable as compared to other plant oils (*Tang et al., 2015a*). The main essential fatty acids in quinoa include linolenic acid and linoleic acid which are metabolized to decosahexaenoic acid and arachidonic acid respectively and are well protected from oxidation by vitamin E and other antioxidants (*McCusker & Grant-Kels, 2010*; *Vega-Gálvez et al., 2010*; *Sharma & Lakhawat, 2017*).

Quinoa contains a rich source of vitamins including vitamin A, vitamin B1/thiamin, vitamin B2/riboflavin, vitamin B3/niacin, vitamin B5/pantothenic acid, vitamin B6/ pyridoxine, vitamin B9/folic acid, vitamin C/ascorbic acid and tocopherols/vitamin E and carotenoids (*Bhargava, Shukla & Ohri, 2006*; *Tang et al., 2015a*). The mineral contents of quinoa include copper, calcium, iron, phosphorus, potassium, magnesium, and zinc (*Bhargava, Shukla & Ohri, 2006*; *Vega-Gálvez et al., 2010*). The bioavailability of iron in quinoa is higher because it contains less amount of phytic acid as it has a strong binding affinity with iron. However, the germination of quinoa seeds activates phytase, which hydrolyzes the complex between phytic acid and minerals. Moreover, the processing of quinoa seeds by soaking, cooking, and fermentation further reduced the phytic acid contents (*Caballero, Trugo & Finglas, 2003*).

The outer seed coat of quinoa consists of saponin, steroidal, and triterpenoids aglycone, with one or more sugar moieties. The sugar moiety includes galactose, glucose, xylose, glucuronic acid, and arabinose (*Yendo et al., 2010*). Common properties possessed by saponin include hemolytic activity in contact with blood cells, foaming capacity in aqueous solution, and complex formation with steroidal and cholesterol components of the plasma membrane. Saponin, therefore, protects plants from insects attack and microbial infection. The formation of saponin is an adaptive strategy in response to abiotic stress and other environmental factors (*Ramakrishna & Ravishankar, 2011*). Synthesis of triterpenoids

takes place *via* an isoprenoid pathway leading to the formation of the triterpenoid backbone such as ursane, dammarane, lupeol or oleanane, and various other enzymes cause several structural modifications such as oxidation, glycosylation, and substitution (*Cammareri et al., 2008*).

Phytosterols present in quinoa seeds are stigmasterols, campesterol, sitosterols, and β sitosterols (*Poveda et al., 2013*). β sitosterol is present in much higher quantities than those of rye, barley, maize, and millet (*Ryan et al., 2007*). Phytosterols show hypocholesterolemic effects in humans and this property demonstrates the similarity between cholesterol and phytosterols. Phytosterols compete with cholesterol for the absorption of cholesterol in the intestine and decrease the production of atherogenic lipoprotein in the intestine and liver (*Marangoni & Poli, 2010*). Quinoa seeds contain the highest amount of phytoecdysteroids in the range of 138 to 570 μg/g in comparison with other edible crops. 20-hydroxyecdysteroids are the most abundant of 13 different phytoecdysteroids isolated from quinoa seeds. Extracts of plants containing phytoecdysteroids have been used as stress reducers, muscle builders, and adaptogens (*Dinan, 2009*).

Phenolics are a diverse class of compounds with a hydroxyl group attached to one aromatic hydrocarbon ring. Because of the presence of the hydroxyl group, phenolic compounds possess antioxidant, anti-inflammatory, antidiabetic, antiobesity, cardioprotective, and anticancer activity (*Harborne & Williams, 2000*; *Da-Silva et al., 2007*; *Kelly, 2011*; *Jeong et al., 2012*). Phenolic acids are associated with pectins of the quinoa cell wall (*Gómez-Caravaca et al., 2011*). The most widely investigated subgroup of polyphenols is flavonoids (*Tsao, 2010*). The presence of daidzein (0.70 to 2.05 mg/100 g) and genistein (0.05 to 0.41 mg/100 g) in quinoa was identified by *Lutz, Martínez & Martínez (2013)*. Isobetanin and betanin are rich in quinoa, also have anti-inflammatory and antioxidant properties. Also, their presence in quinoa gives them their black, yellow, and red color (*Tang et al., 2015b*).

In today's fast-paced community, different technological revolutions generate packaged ready-to-eat products with fast cooking time to deliver health-beneficial phytochemicals. Quinoa is processed with specific technologies to develop targets in specific areas of human health such as weight loss, fitness enhancement, celiac diseases, and other metabolic complications such as hyperlipidemia, obesity, hypertension, and diabetes (*Adetunji et al., 2021*). Quinoa should be formulated and processed. Quinoa undergoes superheated oil treatment to remove saponin and also expending the seed to reduce cooking time for consumers (*Bendevis et al., 2014*). *Rhizopus oligosporus* Saito is used in solid-state fermentation for tempeh productions in quinoa seeds. Tempeh produced in this way contains all essential amino acids with lower isoflavone compounds. Various methods have been developed to concentrate quinoa oils, saponins, phytoecdysteroids, and proteins. Quinoa protein concentrate has been produced *via* hydrolysis and precipitation *via* enzyme or alkali.

## Survey methodology

We conducted a thorough research of the following literature database including Google Scholar, NCBI, Web of Science, Pubmed, Sci-hub, and Scopus. Several keywords and phrases were searched including quinoa, properties, food, cereals, cultivation, geographical distribution, abiotic stress, water stress, soil stress, heat stress, plant environment, quinoa adaptation, factor affecting crops, the factor of adaptation, drought, genetic diversity, genetic response towards drought, response towards abiotic stress environment, salinity stress, antioxidant metabolism, Frost-resistance mechanism, food crop, physiological adaptation, and morphological adaptation. Although we referred to very old publications for key concepts, we largely relied on 20-year publications in our study. To understand the fundamental concepts, we also searched Google images for schematic figures and diagrams. We exclude the studies having abstracts available only with no full-text articles.

## GEOGRAPHICAL DISTRIBUTION

Quinoa was first described as a native species of South America by Wildenow in 1778. According to Buskasov, Peru and Bolivia are the main centers of origin of quinoa (*Cárdenas, 1944*). Gandarillas corroborated its wide range of geographical distribution with a great variety of ecotypes and its economic and social importance both in wild and technically cultivated varieties. The Andean region is considered one of the most important centers of origin of cultivated species (*Arriz et al., 2016*). Quinoa is divided into five categories depending upon agricultural conditions such as Yungas, valleys, salt flats, altiplano, and sea levels with different agronomic, adaptive, and botanical characteristics.

From 7,000 years ago until the beginning of the 1980s, the cultivation of quinoa was only relative for the civilizations within the Andes (*Jacobsen, 2003*). Growing experimentations on the quinoa plant lead researchers in other countries to recognize the benefits and potential of quinoa. Quinoa growth in several countries has increased rapidly. In 1980, the total number of countries growing quinoa species was eight increased to 75 by 2014 with a further increase in countries in 2015 (*Bazile, Jacobsen & Verniau, 2016*). The first experiments were conducted in Kenya in 1935 outside the Andes, with varieties obtained from the Royal Botanical Gardens, UK (*Bazile, Jacobsen & Verniau, 2016*). Different trials were conducted on quinoa resistance to varied environments such as the response in nutrient deficiency conducted in 1948, tolerance in varied temperatures conducted in 1968, and growth response in salinity conducted in 1950. Chilean germplasm was then used by researchers at Colorado State University, USA in 1980. Quinoa was also cultivated commercially in Canada in the same period. After that, quinoa cultivation was introduced in the UK, Tibet, Denmark, India, China, Netherlands, Cuba, and Brazil (*Bazile & Baudron, 2015*; *Quae & Bazile, 2015*).

Food production and distribution to varied environments depend upon the different pillars of food security, such as access to available food and its consumption and utilization. Quinoa due to its genetic variability, nutritional quality, low production costs, and adaptability to varied soil and climate condition is considered a strategic crop that contributes to food sovereignty and security. The main producer of quinoa in the world is Peru and Bolivia (*Quae & Bazile, 2015*). In 2013, the total area under the cultivation for

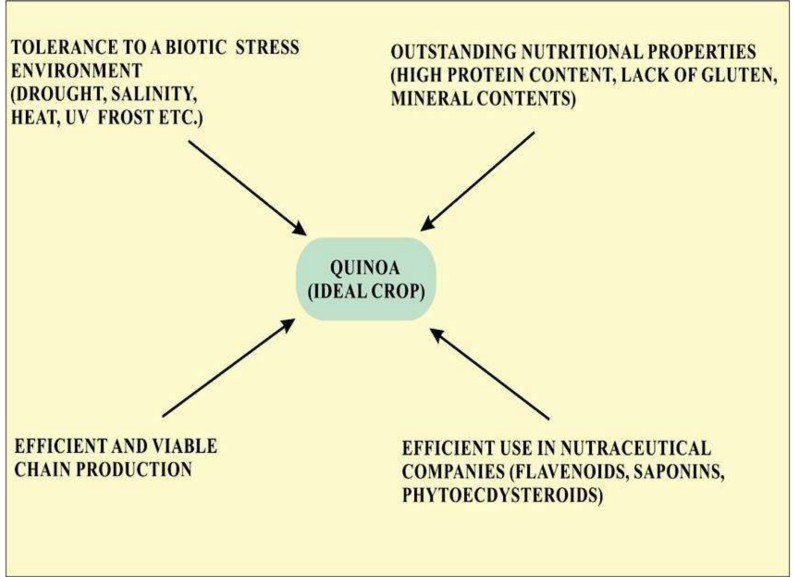

**Figure 1 Properties of quinoa that make it an ideal crop for cultivation.**

quinoa was 45,000 ha in Peru and 75,000 ha in Bolivia. It is estimated that more than 20% of quinoa was produced by these two countries followed by the USA, Chile, China, Ecuador, France, Argentina, Canada, and Ecuador, which together represent about 15–20% of the total world production. The area under cultivation for quinoa plants in Europe was 5,000 ha in 2015 mainly in the UK, Spain, and France. FAO has tested different varieties of quinoa cultivated in different regions outside the Andean region. FAO testing aims to promote resistance to stress, nutritional quality, and food security. Quinoa is cultivated in almost 95 countries of the world now a day (*Quae & Bazile, 2015*).

## QUINOA AS AN IDEAL PLANT FOR CULTIVATION IN MULTIPLE ENVIRONMENTAL CONDITIONS

Quinoa is considered an ideal crop for cultivation in terms of abiotic stress tolerance and is also recognized as the model crop for agricultural production worldwide. Quinoa is used an ideal food crop as a means to fight malnutrition globally because of its outstanding nutritional property (Fig. 1). Screening of various quinoa cultivars revealed their efficiency to use water, tolerance to salinity, and various other attributes leading to growing interest in quinoa cultivation (*Hasegawa, 2013*). Quinoa has been considered a climate-resistant crop with multiple positive attributes, including drought, frost, and salinity resistance and has created a global interest in its cultivation (*Zeglin et al., 2013*). Seeds of quinoa are highly nutritious with high protein contents (methionine, lysine, and threonine), minerals contents (Fe, Ca, and Mg) and vitamins and lack gluten that are limited in legumes and cereals. The United Nations designated 2013 as the year of quinoa because of its exceptional nutritional quality and ability to grow in varied environments (*Bazile et al., 2012*). Knowledge about the effect of extreme conditions on the nutritional and anti-nutritional properties of quinoa will provide information about the introduction of

quinoa in different environmental conditions (*Pulvento et al., 2012*). Quinoa is thought to be an ideal crop for NASA's Controlled Ecological Life support System (CELSS). Criteria on which quinoa is considered a potential crop according to CELSS include its canopy stature, desired nutritional composition, duration of life cycle and harvest index, and high productivity rate (*Schlick & Bubenheim, 1996*). The amino acid profile in quinoa seeds is generally well-balanced. Saponins are present in the pericarp of quinoa seeds which is considered useful for long-term travel. In the international market, the prices of quinoa in the US and European markets per matric tone are five times higher as compared to soya thus providing a promising economic advantage in comparison with other crops thus opening a substantial opportunity for an efficient and viable chain production (*Hossam & Helmy, 2014*).

The main task of research on food production is the safety and efficacy of food production for the growing population under low input management. Today, the scarcity of several resources important for irrigation such as salinity in soils and water resources is the primary cause of loss in crop production worldwide (*Jacobsen et al., 2013*). The remarkable tolerance of quinoa to the hostile environment makes it a suitable candidate crop for facing these challenges to food security. Quinoa may also allow farmers in water shortages and quinoa also grows in salt-affected environments for remediation and revegetation (*Adolf et al., 2012*). Different varieties of quinoa cultivated in environmental conditions outside their traditional growing areas are exposed to different climate conditions and this expansion of cultivation brings about a broad spectrum of diseases and pests.

The use of quality food for the aging and growing population has focused attention on nutraceutical and functional foods. Quinoa accomplishes all the nutritional requirements with properties that promote human health. Quinoa seeds also have antioxidant properties (*Repo-Carrasco-Valencia et al., 2010*; *Vega-Gálvez et al., 2010*). Flavonoids found in quinoa such as quercetin and kaempferol show antioxidant and anticancer properties. Quinoa also contains ecdysteroids that protect plants against nematodes and insects (*Kumpun et al., 2011*). These compounds show strong antioxidant activity and also prevent aging by inhibiting collagenase. The presence of phytoecdysteroids is limited to a few cultivated plants, so quinoa is an important source of such metabolites. Saponins present in quinoa have antibiotic, insecticide, pharmacological, and fungicidal properties (*Ramakrishna & Ravishankar, 2011*).

## Adaptations of quinoa to respond to stress conditions

Quinoa flourishes under a wide range of climate and soil conditions in arid, cold, and wet regions. The adaptability of quinoa in different environmental conditions is due to the differentiation of a variety of ecotypes produced in varied agro-environment (*Pihlanto et al., 2017*). The plant shows various adaptations from physiological to morphological that serve a range of responses to drought and water deficits not only avoidance but also tolerance and resistance. Responses involve the change in root and leaf growth and many cases change in ontogenetic variation. All possible mechanism involves in abiotic stress tolerance involves whole tissue and plant levels including determination from

physiological, molecular, architectural, and morphological perspectives (*Silva et al., 2015*). Quinoa represents the opportunity for both present and future challenges and also serves as an important source of genes with important biotechnological applications.

## DROUGHT

Quinoa is a model crop with the capacity to grow and produce seeds in Chile with semi-desert conditions, arid mountain regions present in northwest Argentina, and the Altiplano region in Bolivia and Peru (*Fathi & Kardoni, 2020*). These areas have less than 200 mm of rainfall and are considered extremely arid (*Hinojosa et al., 2018*; *Fathi & Kardoni, 2020*).

Different drought mechanisms are involved in plants to cope with water shortages. Three different approaches involved are molecular, morphological, and physiological strategies. The presence of aquaporins and activating stress proteins are included in the molecular strategy. The morphological strategy includes the presence of deep roots and scape that are involved in phenotypic flexibility while the physiological strategy involves cell membrane stabilization, osmotic adjustment, stomatal conductance, and antioxidant defense mechanisms (*Hinojosa et al., 2018*).

### Genetic response against drought

Improvement in crop management has drawn the attention of researchers to focus on genetic studies. There was a study conducted by Maugham's group in 2011 that reported the immature seed and floral tissue expressed sequence tag (EST) database for quinoa. These derived sequences underwent homology analysis with known gene and single nucleotide polymorphism (SNP) identification for quinoa (*Maughan et al., 2011*). Furthermore, they compared the 424-cDNA sequences of quinoa with publicly available databases (*Silva et al., 2015*). Almost 67% of quinoa proteins were found homologous of Arabidopsis proteins with reported functions and 9% were significantly homologous of Arabidopsis proteins but with unknown functions. A total of 18% percent did not share any significant homology with the available database however, 6% of proteins were found to be homologous to plant proteins other than Arabidopsis species (*Jarvis, 2006*). The high frequency ESTfound in the quinoa and floral cDNA libraries are Bet v I allergen family, anti-microbial protein 2, lipid transfer protein 3, plant defensing-fusion protein, 60S ribosomal protein L23, and non-specific lipid transfer protein 2. Interestingly, three out of six mentioned EST clusters have putative functions associated to plant defense (*Coles et al., 2005*). Amplification and sequencing of 34 ESTs were performed in five quinoa and one *C. berlandieri*, which identified 51 SNPs in 20 ESTs of quinoa (*Jellen et al., 2013*). Recent works of Maugham not only identified 14,178 SNPs but also reported two subgroups named Andean and coastal quinoa ecotypes from 113 diverse quinoa accessions that were compared with formerly used five accessions (*Maughan et al., 2019*; *Rey & Jarvis, 2021*). Therefore, identified SNPs are considered valuable genomic tools that can help in discovering agronomic traits in quinoa. Many researchers such as *Morales et al. (2011)* also studied early drought stress effects (up to 9 days after sowing) by using Altiplano Chilean

quinoa genotype (*Bazile, Jacobsen & Verniau, 2016*). They analyzed transcriptome in dry and normal irrigation conditions by illumine paired-end sequencing method.

## Physiological adaptation and antioxidant response of quinoa under drought condition

Quinoa's response to low water availability is categorized into stress tolerance and stress avoidance. Mechanisms involved in stress avoidance are the maintenance of water loss and water uptake (*Bandurska, 2022*). Water uptake is enhanced by the accumulation of solutes which decrease the potential of tissues by increasing the growth of roots. Loss of water through evaporation is inhibited by closing the stomata resulting in the restricted growth of the shoot and increased leaf senescence (*Shakeel et al., 2011*). The stress tolerance mechanism is involved in protecting cells from damage when stress becomes more severe. Stress tolerance mechanisms involved the detoxification by solutes, such as proline and late embryogenesis abundant proteins or by using antioxidants (*Ali, Bano & Fazal, 2017*). The involvement of abscisic acid- in both tolerance and avoidance response and abscisic acid- independent mechanism including dehydration response element-binding proteins have also been also extensively elucidated (*Roychoudhury, Paul & Basu, 2013*). Quinoa has a unique ability to cope with drought by resuming former photosynthetic levels in low water levels (*Jacobsen, Liu & Jensen, 2009*; *Silva et al., 2015*). Because of the ability to respond to water shortage, quinoa plant is suitable for growing in semi-arid and arid regions (*Bhargava, Shukla & Ohri, 2006*). Quinoa has a deep root and branch system that penetrates up to 1.5 m in the soil (*Flores, 2012*) and transpiration could be reduced by the presence of calcium oxalate (*Siener et al., 2006*). The plant can also avoid drought by shedding its leaves that reduce surface area, by stomatal regulation, and by the formation of thick-walled cells that preserve turgor (*Jensen et al., 2000*; *Silva et al., 2015*). Precocity, also known as early genotype, is a crucial defense mechanism in regions that might face a water scarcity towards the end of the growth season. It also works through low osmotic potential and the capacity to keep leaves turgor even at extremely low water potential (*Bhargava, Shukla & Ohri, 2006*). Mechanism of modifications of quinoa in drought such as increased level of abscisic acid, rapid closing of stomata, increased osmoprotectants such as proline and betaine, *etc.* are common. However, other mechanisms such as calcium oxalate accumulation, increased thermostability of chlorophyll, and protein stability are not indicated as shown in Fig. 2.

Recent reports suggest that quinoa displays a different mechanism to respond to water shortage as compared to other plants such as maize (*Jacobsen, Liu & Jensen, 2009*). Osmotic adjustment may play a major role in quinoa to maintain turgor under drought conditions (*Jensen et al., 2000*). The specific surface area of leaf and photosynthetic rate during early vegetative growth may cause early vigor that increases tolerance to drought. Stomatal exposure induced by drought may be explained by mechanisms involving the role of cytokinins which are classical antagonists of abscisic acid (*Jacobsen, Liu & Jensen, 2009*). The limited supply of N reduces the cytokinin transport in the xylem, and stomatal sensitivity to abscisic acid may be elevated. So it is concluded that the quinoa plant closes its stomata during soil drying, maintaining leaf potential and rate of photosynthesis that

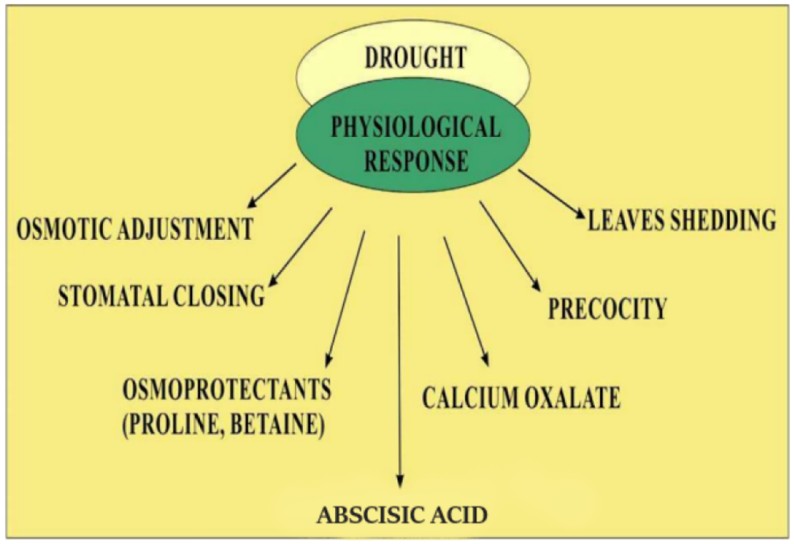

**Figure 2 Physiological response of quinoa in drought stress.**

results in the increased efficiency of water usage (*Jacobsen, Liu & Jensen, 2009*). Increased levels of sugars proline, glycine, and betaine were reported in the quinoa plant during conditions of salt stress (*Ruffino et al., 2010*). These sugars have been recognized as the major osmolyte in different quinoa species. Proline scavenges free radicals, thus preventing membrane protein denaturation during the condition of severe osmotic stress (*Shabala et al., 2012*). Accumulation of proline and other sugar contents maintain the turgor inside the cell that is necessary for the expansion of cells during stress condition Fig. 3 (*Ruiz-Carrasco et al., 2011*).

Biochemical stress induced by a change in water availability may be associated with enhanced production of reactive oxygen species (ROS) in plants which is responsible for oxidative damage. Increased activity of antioxidant enzymes, such as superoxide dismutase (SOD), peroxidase, catalase (CAT), and polyphenol oxidase, was observed in drought conditions with a significant increase by 39% to 90% (*Iqbal et al., 2018*).

## QUINOA RESPONSE UNDER SALINITY CONDITIONS

Halophytes are plants that are adapted to survive under a high concentration of salts and thus represent an ideal crop to understand different genetic and physiological adaptations in salinity stress tolerance. Thus halophyte farming is a potential source of global food production in a progressively salinized world (*Bromham, 2015*; *Flowers & Colmer, 2015*). Quinoa is considered a crop that has the potential to survive in varied salt conditions as shown in Fig. 4.

## GENETIC DIVERSITY

Quinoa is divided into five ecotypes with broad genetic variability that is adapted to the altiplano, valley, sea level, tropics, and salt desert (*Murphy et al., 2016*). The genetic variability of quinoa and its biodiversity is confirmed by using different molecular

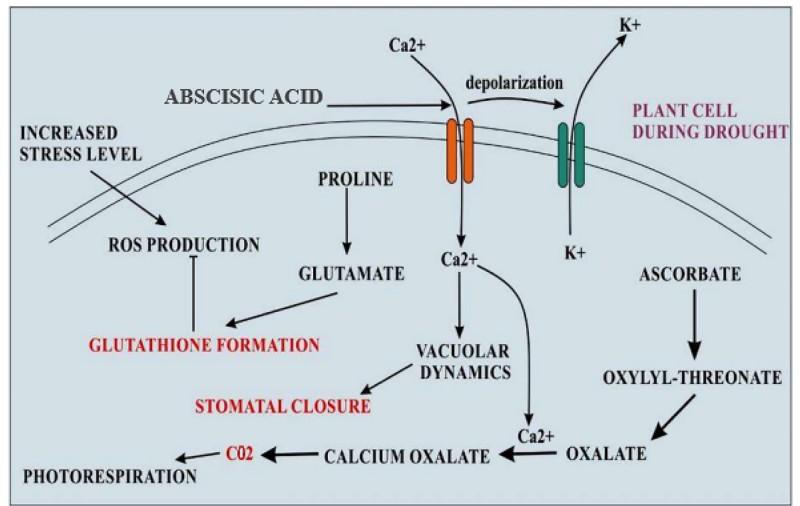

**Figure 3 Cellular adaptations of quinoa in drought stress.** During the condition of drought stress, levels of abscisic acid become increased that resulting in increased calcium influx and K+ efflux by depolarizing the plasma membrane. Increased $Ca^{2+}$ levels inside the cell change the vacuolar dynamics and stomatal closure. The formation of oxalate also becomes increased which becomes the main source of $CO_2$ for photorespiration during stomatal closure. The elevated level of proline accumulation takes that converts into glutamate which is an amino acid used for glutathione formation, a major antioxidant.

**Figure 4 Salt tolerance mechanisms in quinoa plants.**

methods. Comparison of quinoa genotypes under different growth, germination, and yield under varied saline conditions is important to understand the salinity stress tolerance in quinoa (*Hussain, Al-Dakheel & Reigosa, 2018*). Almost 3,000 quinoa accessions are available that show different responses in saline conditions during germination and growth. An observation of *Jacobsen, Mujica & Jensen (2003)* reveals that quinoa can germinate under saline conditions of up to 57 dS m21. The previous study reported that 15 out of 182 Peruvian accessions were shown to be the most tolerant crop with a germination rate of 60% at a salinity level of 25 dS m21 (*Gómez-Pando, Álvarez-Castro & Eguiluz-De La Barra, 2010*). Another study reported the effect of salinity in four Chilean coastal accessions and observed its effect on *in vitro* germination, growth, and other short-term physiological responses (*Ruiz-Carrasco et al., 2011*). The expression level of two Na transporter genes CqNHX salty over sensitive 1 CqSOS1 and their association with these parameters were observed. A significant reduction in the length of roots and germination

rate of seeds was observed in genotype BO78 from southern Chile at the highest NaCl level (300 mM). The lowest level of root/shoot fresh weight ratio was also observed in BO78. The following observations suggested the least salinity-tolerant property of BO78 (*Ruiz-Carrasco et al., 2011*; *Bueno & Cordovilla, 2020*). Moreover, another observation on four Chilean genotypic varieties showed that with a NaCl concentration of 200 mM, Hueque was shown to be the most affected genotype with a 50% decrease in germinability and the lowest decrease in germination rate was observed in Amarilla (*Delatorre-Herrera & Pinto, 2009*). Several mechanisms may contribute towards different genotypic differences in salinity tolerance includes the high exclusion of Na and retention of K ion from leaf mesophyll, increased rate of H pumping, and reduction of vacuolar channels activity under salinity condition. The development of these mechanisms in quinoa causes an increase in the salinity tolerance of quinoa (*Bonales-Alatorre et al., 2013*).

## MORPHOLOGICAL ADAPTATIONS

Several findings have indicated that halophytes can be sensitive to salinity during seed emergence and germination (*Debez et al., 2004*). Understanding the mechanisms involved in the sensitivity and tolerance of seed in the accumulated Na concentration is, therefore, an important concern (*Hasegawa et al., 2000*). It is indicated that the preservation of ion homeostasis is the main mechanism involved. The distribution and maintenance of other ions in tissues and seeds are also important features. A study by *Koyro & Eisa (2008)* studied the mineral distribution of seeds that were taken from plants grown under high salt concentrations. They noticed the altered concentration of salt but highly regulated and did not cause any harm to seeds and also affect the viability of seeds. Although the weight of the seed is decreased at high concentration of salts, reduction in dry matter including carbon-containing compounds that were mainly compensated by increasing ash content (*Koyro & Eisa, 2008*). An increase in ash content induced by high salt concentration was due to high concentrations of Na, K, Mg, and Ca. Therefore, there was a stable buildup of K ions and other essential nutrients including P and S, even under high salinity. Thus seed coat restricted the passage of possibly toxic Na ions and Cl ions to the interior of seeds, as 0.90% Na ion and Cl ions were present inside the seed coat (*Koyro & Eisa, 2008*). The study revealed that the seeds of quinoa plants that were grown under high salt concentration exhibit a tolerance mechanism that was based on the presence of periplasm that serves as a protective barrier and integrity of coat that ensures the exclusion of Na and Cl ions and the maintenance of balanced Na and Cl ion ratios in the interior of seed. Similarly, the viability of the seed depends on its ability to avoid toxicity by excluding Na ions from the developing embryo (*Hariadi et al., 2011*).

A most distinctive characteristic of quinoa is the presence of bladders or salt glands also known as trichomes. Accumulation of the absorbed salts into these bladders may provide an efficient strategy contributing to salinity resistance in quinoa and other salt-tolerant species (*Agarie et al., 2007*; *Ben Hassine et al., 2009*). They are involved in the compartmentalization of accumulated salts and then exclusion from mesophylls. They act as the secondary epidermis and reduce the loss of water and UV-induced damage. In chenopods, these types of structures are present on lower and upper leaf surfaces,

panicles, inflorescences, and stems, and, are also termed as epidermal bladder cells (EBCs). The presence of EBC in rich density in young leaves of quinoa protects Photosystem II (PSII) from UV damage (*Shabala et al., 2012*). Thus, EBCs may accumulate organic compounds with chaperone ability and scavenging of reactive oxygen species (ROS). Quinoa leaves possess calcium oxalate crystals that are related to the accumulation of excessive Ca under salt stress and drought (*Riccardi et al., 2014*). Further studies are required to study the genetic variability of EBC and their specific importance and composition in quinoa.

Saline conditions generally decrease the transpiration rate. The observed reduction in stomatal conductance in halophyte leaves is assumed to be important for better water use efficiency. This may originate from both physiological, *e.g.*, control over stomatal aperture, and morphological, *e.g.*, stomatal density and size, adaptive responses to salinity. A reduction of up to 50% in stomatal density under strongly saline conditions (600,750 mM NaCl) accompanied by reduced stomatal size was reported in the relatively salt-sensitive Chilean cultivar BO78 (*Orsini et al., 2011*). In a comparative study between 14 varieties of quinoa differing in salinity tolerance, *Shabala et al. (2012)* demonstrated that, while all had reduced stomatal density under saline conditions, this morphological parameter was differentially affected in different genotypes.

## INCREASE IN ANTIOXIDANT METABOLISM IN QUINOA UNDER SALT STRESS

Water absorption reduced as a result of high salt concentration directly affects $CO_2$ absorption and stomatal closure that leads to restricted $CO_2$ fixation (*Munns, James & Läuchli, 2006*). Limited $CO_2$ fixation causes a decrease in NADPH oxidation by the Calvin cycle. As a result of that, the electron of NADPH that causes the reduction of NADP+ goes to $O_2$ resulting in the production of excessive ROS. ROS reacts with different macromolecules causing protein denaturation, DNA mutations, and lipid peroxidation (*Gill & Tuteja, 2010*). Plants show different antioxidant responses to minimize ROS-induced tissue damage. Quinoa plants increase the activity of different antioxidant enzymes such as superoxide dismutase (SOD) and catalase (CAT) (*Khalofah, Migdadi & El-Harty, 2021*). SOD catalyzes the conversion of $O_2$.- into $H_2O_2$ and $O_2$ and CAT cause the conversion of $H_2O_2$ into $H_2O$ and $O_2$ (*Gill & Tuteja, 2010*). Moreover, betalains secondary metabolites in the leaves and stems of quinoa also shows antioxidant properties (*de Oliveira Junkes et al., 2019*). Betalains as osmolytes cause the protection of different physiological processes against abiotic stresses (*Wang et al., 2021*). Generally, high salinity causes the accumulation of amino acids such as phenylalanine that act as a precursors for the biosynthesis of betalains (*Tanaka, Sasaki & Ohmiya, 2008*).

A study by *Shabala et al. (2012)* indicated that under different sodium concentrations, the dry weight of roots and shoots of seedlings of quinoa were unaffected. A high level of betacyanin content was reported after treating with 0 to 150 mM of sodium chloride. Moreover, higher activities of antioxidants such as CAT and SOD, were also reported in roots and shoots of early seedlings in 150–200 mM of sodium chloride concentrations. Levels of ascorbate were also higher in shoots under 100–200 mM of sodium chloride

concentration. However, in roots higher ascorbate concentration was reported at 50 mM of sodium chloride. On the other hand, increased lipid peroxidation was observed in shoots under 200 mM of sodium chloride. Nevertheless, no differences were reported in roots between treatments. As far as the levels of $H_2O_2$ are concerned increased levels were observed in shoots under 150–200 mM of sodium chloride. However, in roots, no differences were observed between treatments. Superoxide dismutase is considered as the first line of defense under stress conditions (*Gill & Tuteja, 2010*). Increased SOD activity in high sodium chloride concentration causes the overproduction of $H_2O_2$ that causes toxicity which results in increased catalase (CAT) and ascorbate peroxidase (APX) activity for the detoxification of ROS (*Tang et al., 2015b*). These enzymes work with different affinities for $H_2O_2$. CAT shows a low affinity to $H_2O_2$ and needs a high concentration of its substrate whereas APX exhibits a high affinity to $H_2O_2$ and is capable of detoxifying $H_2O_2$ even at a low concentration of substrate (*Munjal et al., 2019*). Under low concentrations of sodium chloride low levels of $H_2O_2$ were produced but higher APX activity was observed confirming the active role of APX even under lower $H_2O_2$ concentrations (*Sharma et al., 2012*). Under 150–200 mM of sodium chloride concentration, an early seedling of quinoa required the higher activity of CAT to detoxify the $H_2O_2$ produced due to the higher activity of SOD. These findings support the role of CAT in the antioxidant system which mainly shows activity under higher $H_2O_2$ concentrations. This coordinated activity of antioxidant metabolism was able to keep lower lipid peroxidation as control plants until the concentration of sodium chloride increases up to 150 mM (*Sharma et al., 2012*).

## ADAPTATION OF QUINOA IN HEAT STRESS

The markedly elevated temperature during the early growth phase of plants is described as one of the most important abiotic stresses due to abrupt climatic change (*Awasthi, Bhandari & Nayyar, 2015*). Worldwide, a broad range of agricultural losses has been accredited to heat alone or most often in combination with drought (*Suzuki et al., 2014*; *Sehgal et al., 2017*). Heat stress can be defined as an elevated level of air temperature that is above the optimum temperature for plant growth and remains elevated for a while adequate to cause damage and hence decrease plant growth and development (*Wahid et al., 2007*). Plants show different responses to heat stress depending on the duration of temperature and developmental stages of plants (*Driedonks, Rieu & Vriezen, 2016*; *Prasad, Bheemanahalli & Jagadish, 2017*).

Heat causes different morphological, anatomical, phenological, and physiological changes. Morphological changes include the inhibition of root and shoot growth and increased branching of stems. Anatomical changes, such as the reduced size of cells and increased trichome and stomatal densities. In addition to these changes, heat stress also causes certain physiological effects including increased fluidity of the membrane, protein denaturation; instability of cytoskeleton, changes in the rate of photosynthesis and respiration and carbon metabolism enzymes; osmolyte accumulation; mitochondrial and chloroplast enzyme inactivation; changes in salicylic acid, ABA, and ethylene; and increased production of secondary metabolites (*Wahid et al., 2007*; *Bita & Gerats, 2013*). Heat stress prompts the production of oxidative stress by generating ROS, in the same way

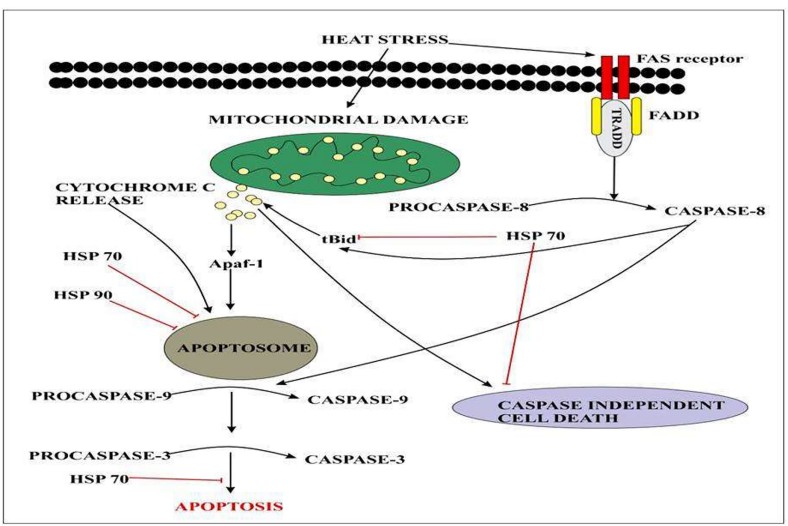

**Figure 5 Extracellular signals of heat stress converge to regulate the mitochondrial damage and caspase activation that will eventually results in apoptosis.**

as in salinity and drought stress which results in serious toxicity in plants; however, ROS in low concentration also acts as a signaling molecule that activates other plants' cellular processes, such as apoptosis (*Hasanuzzaman et al., 2013*).

Heat-shock proteins (HSPs) play a crucial role during gradual or abrupt changes in temperature to cope with the heat stress response (HSR) (*Kotak et al., 2007*; *Wahid et al., 2007*). Several findings have reported that HSPs play an important role in thermo-tolerance in different species of plants. The study indicated that the indispensable role of HSP70 and HSP90 induced tolerance to heat stress response. Heat stress factors (HSFs) are mainly involved in the induction of signaling pathways in HSP expression (*Ohama et al., 2017*). Different genes belonging to heat shock transcription factors were identified. A total of 23 different heat shock transcription factor genes were identified and their expression profiles were identified by RNA seq data. Results indicated that *CqHsfs9 and CqHsfs3* showed higher expression levels after 6 h of heat treatment, while *CqHsfs10 and CqHsfs4* exhibited a higher expression level after 12 h (*Tashi et al., 2018*). Quinoa plants have an excellent heat shock protein network that intervenes caspase cascade at multiple levels Fig. 5.

Several studies have been carried out to indicate the effect of temperature fluctuations on the germination of quinoa (*Mamedi, Afshari & Oveisi, 2017*; *Strenske et al., 2017*). Studies have indicated a positive correlation between temperature and the germination rate of quinoa (*Bois et al., 2006*; *González et al., 2017*). Results suggested that the maximal temperature at which Quinoa germinate is 50 °C, and the minimum germination temperature is 3 °C (*Jacobsen & Bach, 1998*; *González et al., 2017*). In contrast, *Bois et al. (2006)* reported germination temperature fluctuations in 10 different varieties that vary between −1.9 °C and 0.2 °C. Another study reported that the optimum temperature range in three different varieties of quinoa was 18–36 °C however; the maximum and base

temperature range was 54.0 °C and 1.0 °C respectively. Seeds of the quinoa plant can be stored at 25 °C for up to 430 days under controlled environmental conditions, before germination fully declines (*Strenske et al., 2017*).

## Frost-resistance mechanisms

Frost stress occurs in different regions of the world such as Northern Europe, Asia, North America, Africa, and Europe during the winter season. Frost hardness can be defined as the composites of different stress tolerances such as heaving, flooding, snow molds, desiccation, and freezing. However, frost resistance has been thought to be the primary factor in many regions of the world (*Macháčková, McKersie & Leshem, 1995*).

In several parts of the Andean region, tolerance to freezing temperature is considered as the limiting factor for the production of crops. 20–40% reduction in the yield of crop takes place in freezing temperatures, however, in rare cases it will result in the total loss of crop (*Jacobsen, Stølen & Mujica, 1997*). Quinoa is considered an ideal crop for cultivation that can tolerate freezing temperatures to some extent. Quinoa exhibits different mechanisms that prevent the crop from immediate destruction at extremely low temperatures of up to 5 °C. The main mechanism includes the formation of ice in cell walls and then subsequent dehydration without any kind of irreversible damage. The presence of soluble sugars in the quinoa plant such as sucrose, may cause a reduction in the mean lethal temperature (TL50) and implies a high level of tolerance to freezing temperature (*Jacobsen et al., 2007*).

Comparison of different phenological stages of quinoa may indicate excellent tolerance in different freezing temperatures. Quinoa at the two-leaf stage can tolerate −4 °C for up to 4 h with only a slight effect on seed yield. It was estimated that only 9.2% of seed yield was affected in comparison with the control that was grown at 19 °C (*Jacobsen et al., 2005*). However, yield decreased to 50.7 and 65.7% at −4 °C during the 12 leaf stage and flowering stage respectively (*Jacobsen et al., 2007*).

## Ultraviolet B (UV-B) radiation

A small fraction of the solar spectrum may represent ultraviolet B radiation (280–315 nm). However, its high energy can be harmful to living organisms (*Müller-Xing, Xing & Goodrich, 2014*). Tolerance of UV-B radiation in plants depends upon their species origin (*Lindroth et al., 2000*), circadian rhythms (*Horak & Farré, 2015*), and age (*Yao & Liu, 2007*). The discovery of the UVR8 photoreceptor led scientists to reveal whether UV-B is a morphogenetic factor considered as abiotic stress (*Rizzini et al., 2011*; *Jenkins, 2014*). Several studies have indicated the effect of UV-B on quinoa that was grown at higher altitudes in South American countries (*McKenzie, Liley & Björn, 2009*). *Palenque et al. (1997)* described several different responses in pigment synthesis and morphology in three different quinoa varieties such as Sayana, Robura, and Chucapaca. Treatment of quinoa directly to UV-B radiations causes a reduction in leaf size and plant height and also found in flavonoid content in leaf. The best adaptation to UV-B is found in Chucapaca. Previous findings discovered the effects of UV-B in the metabolism of quinoa seedlings at ultrastructural levels. The organization of thylakoid is also affected in response to UV-B exposure (*Sircelj et al., 2002*).

Another study by *Hilal et al. (2004)* indicated that exposure of quinoa plants to UV-B radiations causes the deposition of lignin in the epidermis of quinoa's cotyledons. Enhanced exposure to UV-B radiations causes an increase in leaf content of lignin that may be an effective UV screener. UV-induced lignin accumulation may be coincident with the activity of the peroxidase enzyme involved in lignification processes. Moreover, the study on two altiplano varieties Cristalina and Chucapaca in a semi-controlled experimental condition demonstrated different distribution patterns of glucose, fructose, and sucrose in leaves and cotyledons after exposure to UV-B radiations. An increase in fructose level might be linked to the result of increased activity of the pentose phosphate pathway that increases the supply of erthyrose-4-phosphate that serves as a substrate for the synthesis of phenolic compounds and lignin through the Shikimate pathway. Fructose phosphate and erthyrose-4-phosphate are produced in the same reaction of the pentose phosphate pathway catalyzed by transaldolase. These studies are beneficial to recognize the plasticity of metabolic pathways that are involved in quinoa's tolerance to UV-B radiation. Another finding indicated different morphological changes such as stem diameter, plant height, leaf number, and area in five different varieties of quinoa because of UV radiations under controlled conditions (*Perez, González & Prado, 2015*).

The study by *Prado, Perez & González (2016)* demonstrated the effect of UV radiations on UV-absorbing compounds, photosynthetic pigments (carotenoids, chlorophyll a, b, and total chlorophyll), and soluble sugars (sucrose, glucose, and fructose) in five different varieties of quinoa grown in different geographical. An elevated level of UV absorbing compounds was observed in five varieties that protect photosynthetic apparatus from excessive radiations by acting as a chemical shield. Recently, a study by *Reyes et al. (2018)* reported the effect of different levels and duration of UV-B radiation on photosynthesis, synthesis of pigment, chlorophyll, and production of ROS. It is concluded that quinoa can activate different mechanisms, depending on the dosage of UV-B irradiation. Additional studies are necessary to fully elucidate the effect of UV-B radiation such as the exact dosage of UV-B which ceases normal morphogenetic processes and becomes a factor of stress and, relationship between radiations and other environmental factors.

## Quinoa response to heavy metals

Lead is considered as a persistent toxic pollutant because of rapidly increasing anthropogenic stress on the environment (*Pourrut et al., 2011*). Excessive accumulation of lead may cause various physiological, morphological, and biochemical changes in plants (*Shahid et al., 2013*). A study by *Haseeb et al. (2022)* reported a reduction in dry and fresh weight of roots and shoots and root length of four different quinoa varieties in the lead concentration of 100 mg/kg as compared to 50 mg/kg lead concentration. The results revealed that a 100 mg/kg concentration of lead did not stop the growth of all quinoa lines, however; the dry biomass of the plant was affected. When the plant was exposed to 100 mg/kg of lead concentration at the panicle emergence stage significantly less plant height was observed in all quinoa lines.

Lead accumulation inhibits the development and growth of plants due to poor uptake of essential minerals (*Gopal & Rizvi, 2008*). Accumulation of heavy metals in plants is of great

concern due to the risk of food contamination through the interface between soil and root (*Mukhtar et al., 2010*). Absorption of lead from soil solution in plants usually takes place through roots *via* the apoplast pathway or through $Ca^{2+}$ permeable channels (*Sharma & Dubey, 2005*). Bioconcentration factor (BCF) is defined as the ratio between the accumulated metal ion in plant roots to the concentration of metals in soil (*Zhuang et al., 2007*). Less than 1 BCF value was detected in all quinoa varieties. On the other hand, the translocation factor is the ratio between accumulated metal concentrations in shoots of plants to the metal concentration in plant roots. It is estimated that during the reproductive stage, lead taken up by quinoa varieties was not translocated to other parts of plants. This is a unique property present in quinoa however, other plants require external adjustments to reduce the sequestration of metals at the seedling stage and stop its transfer to the flowering stage by immobilizing lead through biochar and decreasing its bioavailability as in rice plants (*Li et al., 2016*). Lead accumulated in high concentrations in plants may block the translocation of other essential nutrients such as Ca, Mg, P, Cu, K, and Zn resulting in the deficiency of these ions. Biochar from animal and plant origin as well as compost influence positively on chemical and biological properties of Ni-contaminated and salt-affected soils. Biochar increases the bioavailability of essential nutrients by decreasing metal toxicity and improving water retention (Fig. 6). Lead toxicity may lead to the reduction in the contents of carotenoids and chlorophylls a and b in mung beans (*Deshna & Bafna, 2013*) and wheat (*Lamhamdi et al., 2013*). The results of this study were similar to the study of *Akinci, Akinci & Yilmaz, 2010* who indicated the destruction of the thylakoid membrane and grana in lead toxicity. In quinoa plants, lead at high concentrations may destroy chlorophyll A and b. Overproduction of reactive oxygen species may lead to the disruption of the chloroplast which is the main cause of reduction in photosynthetic pigments (*Pourrut et al., 2011*). Increased oxidative stress due to lead toxicity may lead to the destruction of chlorophyll in rice plants (*Zeng et al., 2007*). Under abiotic stress conditions, carotenoids work as antioxidants in addition to photosynthesis (*Kasote et al., 2015*). Under high lead concentration in quinoa lines, increase carotenoid concentration was reported as compared to chlorophyll. High carotenoid concentration may play an important role in quenching free radicals produced during stress conditions under lead toxicity (*Kong et al., 2015*). Different defense strategies are used to cope with lead toxicity. Such strategies include reduced lead uptake into the cell; sequestration of lead into vacuoles by the formation of complexes; binding of lead by phytochelatins, glutathione, and synthesis of osmolytes (*Pourrut et al., 2011*). Phenolic compounds synthesized in the plants prevents the diffusion of free radicals and stabilize the membrane. Phenolic compounds make hydrogen bonds with polar heads of phospholipids of present in membranes (*Michalak, 2006*). Quinoa plants exhibited a high concentration of phenolic compounds in lead toxicity. *Zeng et al. (2007)* described that the accumulation of lead negatively affected the sugar beet *(Beta vulgaris)* yield.

## Biotic stress in quinoa

Plant disease susceptibility is one of the major biotic factor in impacting the growth and yield of plants. *Peronospora variabilis* causes downy mildew, one of the most important

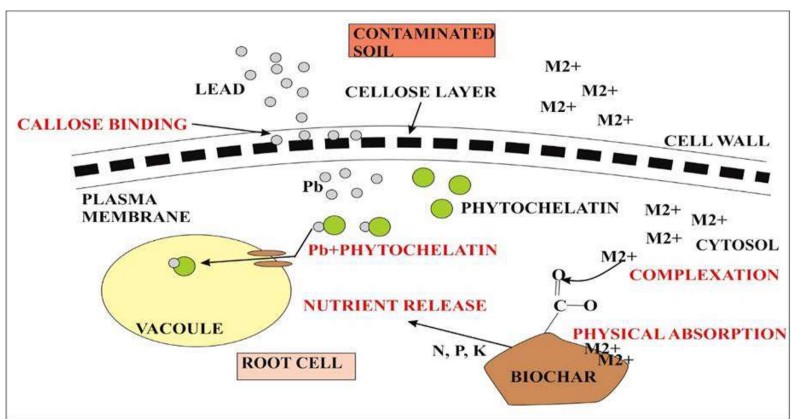

**Figure 6 Representation of quinoa root response during metal toxicity.** The first defense in quinoa is the presence of callose between the cell wall and plasma membrane. Callose binds with the metal lead and prevents its accumulation inside the cytosol. Callose inhibits cell-to-cell transport. The second defense is the chelation of $Pb^{2+}$ with phytochelatins. $Pb^{2+}$ complex will eventually take up by vacuoles that are formed at an additional level during metal toxicity. On the other hand, metal ions ($M^{2+}$) toxicity inside the cytosol can also reduce by biochar accumulation. $M^{2+}$ forms complex with functional groups present on the surface of biochar or it can also be absorbed physically by biochar itself. Biochar also causes nutrient release for plant root cells.    

diseases affecting quinoa yield, with the yield losses of upto 99% for susceptible varieties and 33% for tolerant varieties (*Bhargava, Shukla & Ohri, 2006*; *Testen, McKemy & Backman, 2013*). *P. variabilis* was formlly classified as endemic in south America. In 1990, this pathogen was identified causing disease on quinoa in Canada and afterwards in the United States in 2012 (*Tewari & Boyetchko, 1990*; *Testen, McKemy & Backman, 2013*). Besides downy wildew, numerous other diseases have been observed globally and are known to drastically decline yield and growth of quinoa. Two diseases related to stalk and leaf of quinoa are leaf spot and brown stalk rot caused by *Ascochyta hyalospora* and *Phoma exigua var. foveata Boerema*, respectively (*Danielsen, Bonifacio & Ames, 2003*; *Mathur & Kongsdal, 2003*). *A. hyalospora* was discovered on quinoa in Pennsylvania in 2011 (*Testen, McKemy & Backman, 2013*). It is initiated by reddish-brown foliar lesions that progress to circular necrotic areas with black asexual fruiting bodies called pycnidia dispersed inside the lesion; however, *Phoma exigua* is a soil-borne disease that prefers low temperature and high humidity. The presence of pycnidia is a diagnostic marker of this disease, and important symptoms include chlorotic leaves and downward bending stems that break readily. It has also been reported to infect other crops such as potatoes. For host invasion and infection, the *P. exigua* requires natural openings, leads to recuerent outbreaks in the Andes after hailstorms (*Danielsen, Bonifacio & Ames, 2003*). In 1998, researchers at Lima's International Potato Center identified *Rhizoctonia solani* and *Fusarium species* from quinoa (*Gleń-Karolczyk, Witkowicz & Elżbieta, 2016*). *R. solani* cause damping off symptoms *i.e.*, seed germination failure andsunken stem leasions. Wilting and root rot is associated with *Fusarium spp. Sclertoium rolfsii Sacc.* discovered on quinoa in 1980, is related to seed rot and damping off. Plant collapse and stem girdling are indications of *S. rolfsii* infection (*Danielsen, Bonifacio & Ames, 2003*; *Gleń-Karolczyk, Witkowicz &*

*Elżbieta, 2016*). Pythium zingiberum inoculation on soil cause damping off to susceptible quinoa seeds (*Ikeda & Ichitani, 1985*).

Numerous plant viruses have been reported in quinoa (*Walkey, 2012*). The leaves of quinoa regularly develop lesions infected with groundnut chlorotic fan-spot tospovirus (GCFSV), 4 days after inoculation (*Chou et al., 2017*). The *quinoa* is susceptible to the viruses infected the host plant that grow next to it. The most commonly known viruses are sowbane mosaic virus, amaranthus leaf mottle virus, arracacha virus, ullucus mild mottle virus, potato virus S, potato andean latent virus, cucumber mosaic virus, passiflora latent virus, plantago asiatica mosaic virus, and carnation latent virus (*Hollings & Stone, 1965*; *Gibbs et al., 1966*; *Dias & Waterworth, 1967*; *Bos & Rubio-Huertos, 1971*; *Brunt et al., 1982*; *Herrera, Juárez & Muñiz, 2003*; *Segundo et al., 2007*; *Hammond, Bampi & Reinsel, 2015*).

## CONCLUSION AND FUTURE PERSPECTIVES

The main challenge of today's world is to meet fiber needs and future food requirements which was more discouraging than experienced under the green revolution. The primary growth-stimulating factors are chemical fertilizers and irrigation water which cannot be expected to increase yield at the required rate. About one-third of the world's land surface is facing drought, salinity, extreme variation in temperature, and many other abiotic problems due to an increase in global warming. Quinoa is the only crop that has resistance to adverse environmental factors. The improved knowledge of mechanisms involved in the resistance of quinoa to adverse abiotic and biotic stress environments will help to overcome the limitations imposed by these stresses all over the world. There are expectations of increased commercial production of quinoa so there is a need for time to regulate the international framework on different genetic resources that are needed to facilitate the breeding of quinoa plants in varied environmental conditions.

### Funding

The Deanship of Scientific Research at Umm Al-Qura University supported this work by Grant Code: 22UQU4350073DSR12. The funders had no role in study design, data collection and analysis, decision to publish, or preparation of the manuscript.

### Grant Disclosures

The following grant information was disclosed by the authors:
Umm Al-Qura University: 22UQU4350073DSR12.

### Competing Interests

The authors declare that they have no competing interests.

### Author Contributions

- Qura Tul Ain conceived and designed the experiments, performed the experiments, prepared figures and/or tables, and approved the final draft.

- Kiran Siddique conceived and designed the experiments, performed the experiments, prepared figures and/or tables, and approved the final draft.
- Sami Bawazeer analyzed the data, prepared figures and/or tables, and approved the final draft.
- Iftikhar Ali analyzed the data, prepared figures and/or tables, and approved the final draft.
- Maham Mazhar conceived and designed the experiments, performed the experiments, analyzed the data, authored or reviewed drafts of the article, and approved the final draft.
- Rabia Rasool conceived and designed the experiments, performed the experiments, prepared figures and/or tables, authored or reviewed drafts of the article, and approved the final draft.
- Bismillah Mubeen conceived and designed the experiments, performed the experiments, analyzed the data, prepared figures and/or tables, authored or reviewed drafts of the article, and approved the final draft.
- Farman Ullah analyzed the data, prepared figures and/or tables, and approved the final draft.
- Ahsanullah Unar analyzed the data, prepared figures and/or tables, and approved the final draft.
- Tassadaq Hussain Jafar conceived and designed the experiments, analyzed the data, authored or reviewed drafts of the article, and approved the final draft.

## Data Availability

This article is a literature review.

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
