# Peer review of "Adaptive mechanisms in quinoa for coping in stressful environments: an update"

_PeerJ, doi:10.7717/peerj.14832_

## Round 0.1 · original submission · Major Revisions

After reading through the comments, I recommend a thorough revision of the manuscript and resubmit at the earliest.

·

Basic reporting

The manuscript #77341 entitled ‘Adaptive mechanisms in quinoa for coping in stressful environment: An update’ provides a comprehensive review for quinoa research. Basically, the manuscript is written logically and well organized. However, many writing problems can be found in the text and noted below. I strongly recommend that the manuscript must be thoroughly revised for acceptance. Here are my main concerns:

1. Writing problems are everywhere. For example, line 55 should be ‘Quinoa is morphologically….’ and line 65 should be ‘…pancakes and tortillas’. The scientific name of quinoa must be italicized (lines 14 and 50), but the first character of ‘quinoa’ should not be capitalized in the sentence. I won’t list them all here. The English writing of the manuscript must be carefully and thoroughly checked by professionals.
2. The citation format in the main text and in the reference list must be corrected. Authors may refer to the manuscript guidelines to comply with the journal’s requirements.
3. Repeated statements or words should be avoided as much as possible. Some duplicate statements can be considered for deletion. For example, lines 68-69 ‘Quinoa is a unique and culturally important stress-tolerant crop with different phytochemical composition and high nutritional value.’ sounds similar to lines 52-53 ‘Quinoa is also known as a superfood because of its super nutritious quality and stress-tolerant properties.’. The sentences on lines 217-229 and 686-689 sound redundant. The sentence on lines 593-596 is repeated. By the way, the statements on lines 584-589 and 590-597 should cite references.
4. Writing for chemical elements, molecules and compounds requires revision. I recommend giving full element and compound names the first time they are mentioned so that their symbols and formulas can be used later. It also applies to terms, including EST (line 291), SNP (line 292), ROS (lines 450, 469-470 and 676), CAT (lines 366, 473, 475, 486 and 496), SOD (line 495), etc., provide full names in the first description and use acronyms later.
5. In addition, pay attention to the correctness of the chemical formulas. For example, Ions should be superscripted, and numbers should be subscripted in molecules. In Fig. 3, what is ‘Co2”?
6. According to the ‘genetic response against drought’ studies (lines 289-309), what are the significant findings or which genes are associated with drought tolerance?
7. The paragraph on lines 510-518 sounds pointless here.

Experimental design

No comment.

Validity of the findings

No comment.

Additional comments

This review provides a comprehensive overview of abiotic stress responses in quinoa, but does not describe biotic stress responses. For the completeness of this review, I recommend that disease resistance studies on quinoa can be included. Authors can refer to Chou et al., PLoS ONE 2017, 12:e0182425 and Colque-Little et al., BMC Plant Biology 2021, 21:41.

Reviewer 2 ·

Basic reporting

The English language should be improved to ensure that an international audience can clearly understand your text. Some examples where the language could be improved. Some examples., Lines 17, 20, 148, 149, 467, 469, 475 etc I suggest you get a help from professional English reader.

The review is within the scope of the journal

Experimental design

The survey methodology is consistent with relevant references..

Validity of the findings

N.A

Additional comments

The article “Adaptive mechanisms in quinoa for coping in the stressful environment: An update”.
Authors: Qura Tul Ain., Kiran Siddique., Rabia Rasool., Bismillah Mubeen., Maham Mazhar., Tassadaq Hussain Jafar
General comments:
The review article gives a brief overview of the various physiological, morphological and metabolic adaptive mechanisms in Quinoa. The article is written with detailed information. However, there are some issues to be considered.
1) The English language should be improved to ensure that an international audience can clearly understand your text. Some examples where the language could be improved. Some examples., Lines 17, 20, 148, 149, 467, 469, 475 etc I suggest you get a help from professional English reader.
2) Introduction should be rewritten to improve the language. There are many grammatical and spelling mistakes in the text.
3) Figure title should be clearer. In figure 6, Callose or cellulose?
4) Even though the mechanisms are mentioned in the figures, it will be better to explain the mechanism in the text.
5) Literature well referenced & relevant. But the reference citations in the text are not correct. Eg. Line 84, 105, 109, 111,112……..
2. Please check the reference format. Some references are not formatted accordingly.

Reviewer 3 ·

Basic reporting

The authors have done a very good job in providing excellent review entitled "Adaptive mechanisms in quinoa for coping in stressful environment: An update" .It consisted of abstract,introduction,survey methodology,,geographical distribution,its cultivation in multiple environmental conditions,its adaptations to respond to various stress conditions,followed by conclusion and future perspectives,

Experimental design

The authors have conducted a thorough research of the following literature database which include {A.}Google Scholar {B.}, NCBI, Web of Science, {C.} Pubmed, Sci-hub and Scopus.
{D.}The authors have used several keywords and phrases such as Quinoa, properties, food, cereals, cultivation, geographical distribution, Abiotic stress, Water stress, soil stress, heat stress, plant environment, Quinoa adaptation, factor affecting crops, factor of adaptation, drought, genetic diversity, genetic response towards drought, response towards abiotic stress environment, salinity stress, antioxidant metabolism, Frost-resistence mechanism, food crop, physiological adaptation, morphological adaptation.
{E.} The authors have refereed to very old publications for key concepts and largely dependent on 20-years publications in order to understand the fundamental concepts, they also searched
Google images for schematic figures and diagrams.
{F.} They excluded the studies not having full text articles in spite of having abstracts

Validity of the findings

Language corrections need to be done at several places.
Abstract-
Seeds of quinoa have an superb balance ->should be corrected as - Seeds of quinova have a superb balance
climate variations--> Should be corrected as---> climatic variations
which will impact on reliable and safe production of food->Should be corrected as-->Which will have an impact on reliable and safe production of food.
Lines 84,85 should be modified The starch mainly consists of maltose, D-xylose with low
fructose, and glucose content.----> Starch mainly consists of glucose with small amounts of maltose,D-xylose and fructose.
Lines 91 and 92.The monosaccharides composition of quinoa is comparable to that of vegetables, legumes, and fruits -----> This sentence can be removed as it does not convey anything specific and no reference has been given given.
line 94 -remove subunit
lines 150 and 151----> needs to be reframed.---->Isobetanin and betanin are the most abundant betanin with anti-inflammatory and antioxidant effects similar to that of phenolic compounds (Tang, Li, Zhang et al., 2015).
lines 588-589 The presence of soluble sugars in the quinoa plant such as dehydrin, sucrose and fructans may cause a reduction in the mean lethal temperature (TL50) and implies a high
level of tolerance to freezing temperature---->The authors should rewrite the sentence since fructans are not sugars,but polymers.Dehydrins (DHNs) are the proteins that are quite effective in adaptive defense of a plant for salinity, drought, and low temperature stress

Additional comments

English language and references should be doubly checked before it is published.

---

## Round 0.2 · accepted · Accept

The paper is revised as per the suggestions of reviewers and is accepted for publication.

Please see the attached file from Reviewer 1 and the suggested edits from the Section Editor below:

LINE 58 : / considered / considered a / [.]
LINE 59: / rice, however, / rice; however, / [.]
LINE 124: / possess / possessed by / [.]
LINE 175: / on 20-year / on a 20-year span of / [.]

·

Basic reporting

Although the authors revised the manuscript, many writing problems are still found in the main text. The revised manuscript is attached for the authors’ reference. My comments are also included in the attachment. Spelling mistakes in chemical names in figures must be corrected.

Experimental design

no comment

Validity of the findings

no comment

Additional comments

no comment

Reviewer 3 ·

Basic reporting

The authors have conducted a thorough research of the following literature database which include {A.}, Google Scholar {B.}, NCBI, Web of Science, {C.}, PubMed, Sci-hub and Scopus.
{D.}The authors have used several keywords and phrases such as Quinoa, properties, food, cereals, cultivation, geographical distribution, Abiotic stress, Water stress, soil stress, heat stress, plant environment, Quinoa adaptation, factor affecting crops, factor of adaptation, drought, genetic diversity, genetic response towards drought, response towards abiotic stress environment, salinity stress, antioxidant metabolism, Frost-resistance mechanism, food crop, physiological adaptation, morphological adaptation.
{E.}, The authors have refereed to very old publications for key concepts and largely dependent on 20-years publications in order to understand the fundamental concepts, they also searched
Google images for schematic figures and diagrams.
{F.}, They excluded the studies not having full text articles in spite of having abstracts

Experimental design

It consisted of abstract, introduction, survey methodology, geographical distribution, its cultivation in multiple environmental conditions, its adaptations to respond to various stress conditions, followed by conclusion and future perspectives,

Validity of the findings

This has already been provided in the comments.

Additional comments

Nil.This has already been provided.The authors have corrected/modified the manuscript as per the suggestions of the three reviewers.